# Research on the Historical Dynamics of Baicheng Oil Chicken Populations

**DOI:** 10.3390/ani15131952

**Published:** 2025-07-02

**Authors:** Huie Wang, Tianci Liu, Gang Wang, Xiurong Zhao, Chengqian Wang, Fugui Li, Gemingguli Muhatai, Lujiang Qu

**Affiliations:** 1College of Animal Science and Technology, Tarim University, Alar 843300, China; whedky@126.com (H.W.); 13633380110@163.com (T.L.); wcqdky@126.com (C.W.); lifuguiyn@163.com (F.L.); gmgl-113@foxmail.com (G.M.); 2College of Animal Science and Technology, Agricultural University, Beijing 100193, China; wanggang@cau.edu.cn (G.W.); zxiurong_feign@163.com (X.Z.)

**Keywords:** Baicheng Oil Chicken, effective population size, introgression, genetic diversity

## Abstract

This study analyzed the whole genome resequencing data of 162 chickens from 16 breeds to assess the population historical dynamics of Baicheng Oil Chicken (BCY). The main findings include: the historically effective population size (Ne) is approximately 46,066. It differentiated 428 to 548 years ago. Approximately 7% of the genetic infiltration comes from the white-broked pheaser (LH), including functional genes (CTNNAL1 responsible for egg production, RARX responsible for fat deposition). The research results clarified the population dynamics and gene flow of BCY, providing a basis for its protection and utilization.

## 1. Introduction

Baicheng Oil Chicken originated in Baicheng County, Aksu Prefecture, Xinjiang Uygur Autonomous Region—a region historically significant as the cradle of the ancient Kucha culture. The breed has been reared since the Qing Dynasty, with an approximate 300-year domestication history. Traditionally, BCY was maintained under free-range management systems. However, since the 20th century, the introduction of exotic breeds coupled with limited conservation awareness among local farmers has led to extensive genetic introgression. Consequently, both the population size and breeding distribution of BCY have experienced sustained decline, rendering the breed endangered.

In order to protect this local breed, the local government established the Baicheng Oil Chicken Resource Conservation Base in 2007, starting with a foundational population of only 200 individuals. In September 2009, BCY was officially recognized by the National Committee for Livestock and Poultry Genetic Resources [1], and on 15 January 2010, it was listed in the “National Catalogue of Livestock and Poultry Genetic Resources” (Ministry of Agriculture Announcement No.1325). To enhance genetic restoration, local authorities organized a systematic collection effort in 2012, gathering over 600 individuals from rural households that matched the original breed standards in morphology production performance and adaptability. This initiative marked the beginning of structured conservation, purification and breed rejuvenation programs.

According to the Third National Livestock Genetic Resource Census, the current BCY population has rebounded to approximately 363,000 individuals. However, despite centuries of genetic drift, natural selection, and artificial selection, no comprehensive studies have yet investigated the genomic diversity, structural variations, or historical hybridization patterns in this breed.

Introduced by Sewall Wright (1931) in *Evolution in Mendelian Populations*, *Ne* is a fundamental concept in population genetics, quantifying the rate of genetic drift and neutral variation in finite populations [2]. The equilibrium level of neutral or weakly selected polymorphisms scales directly with *Ne*. Secondly, the fixation of beneficial alleles and purging of deleterious mutations depend on the product of *Ne* and selection coefficient. Therefore, *Ne* modulates both the variability of DNA sequences and the divergence rates of coding and non-coding regions [3].

Population genetic studies have been instrumental in unraveling the historical trajectories of various breeds, clarifying their origins and differentiation and supporting genetic breeding efforts. This approach has been widely and successfully applied across numerous domesticated mammals and poultry species, such as chicken breeds [4,5], sheep [6], and pig populations [7], as demonstrated in previous research.

In this study, the whole genomes of 162 chickens from 12 Chinese local breeds, 3 commercial breeds and 1 RJF were sequenced (Figure 1). Thirty Baicheng Oil Chickens were selected as the selection group, and 15 other breeds were selected as the background groups. Then, we performed comprehensive analyses of linkage disequilibrium (LD) decay, demographic history and differentiation time according to their genomic data. We also identified introgression genes from commercial chicken to BCY.

## 2. Materials and Methods

### 2.1. Animals and Samples

This study collected whole-genome resequencing data from 162 individuals across 16 chicken breeds. Based on the existing dataset we studied and collected [8], two additional local breeds—Turpan cockfighting (TLF, 8 individuals) and Luxi cockfighting (LX, 12 individuals)—and one commercial breed, White Rock Chicken (WRR, 6 individuals), were included. All samples were obtained using the Illumina high-throughput sequencing platform, with a sequencing depth of at least 10X The geographical locations of the local chicken breed samples are shown in Figure 1.

### 2.2. LD Decay Linkage Disequilibrium

The population polymorphism was analyzed using VCFtools (v0.1.16) with a 50 kb sliding window. Pairwise SNP r^2^ values were then calculated via the PopLDdecay (v 3.41) [9] software. Lastly, average r^2^ values for SNP pairs within a specific distance range were statistically analyzed using a Perl script.

### 2.3. Estimation of Historical Effective Population Size and Divergence Time

*Ne* of BCY population was clarified using the Hidden Markov Model (HMM) method with SMC++ (v1.15.4) [10]. The divergence time was estimated using fsc27. SNPs in intergenic regions were converted to SFS using easySFS. The parameters included a neutral mutation rate (μ) of 1.9 × 10^−9^ per base per generation and a generation interval (g) of 1 year based on the chicken genome. The fastsimcoal2 (v2.6.0.3) [11] software was run 100 times with different starting points, retaining the model with the maximum likelihood. Each parameter file included 100,000 simulations (-n 100,000) and 40 cycles of Expectation/Conditional Maximization (-L40).

### 2.4. TreeMix Analysis

TreeMix (v1.13) [12] was used to analyze genetic migration events among BCY, 11 local chicken breeds, and 3 commercial chicken populations. The maximum likelihood (ML) tree was constructed with migration event parameters ranging from M = 3 to 15, using 1000 randomly selected SNPs (-k 1000) for variance–covariance calculation. Red Junglefowl served as the outgroup. Model residuals were calculated to evaluate fit, and results were visualized using R (v4.41) with custom scripts.

### 2.5. D Statistical Genetic Infiltration Analysis

Dsuite’s Dtrios [13] module calculated D values and f4-ratio statistics for various population combinations. Dinvestigate further analyzed trios with significantly elevated D values. The genome window size was set to 2500 SNPs with a 500 SNP sliding step. Within these windows, D, f_d, and f_dM statistics were calculated. The top 10% of windows of the highest D statistic were extracted to identify introgression chromosomal locations and introgressed genes. These genes were subjected to GO and KEGG enrichment analysis using DAVID, with significance enrichment determined at *p* < 0.05.

### 2.6. Identification of Candidate Regions for Gene Infiltration

To identify introgression regions and degree in BCY population, this study first screened introgressing breeds using D statistics. Then, Weir and Cockerham’s F*_ST_* estimates were calculated for each individual relative to the introgressing population using VCFtools (v0.1.16) with a 100 kb sliding window and 50 kb step size. The identified introgression regions and degree among individuals in the BCY breed provided positional information and genes. An ML tree of the candidate genes was constructed using IQ-TREE, and genetic variations were annotated.

## 3. Result

After quality control filtering of the whole-genome resequencing data of the 162 samples in this study, 1824.768 Gb of Clean Base was obtained. On average, approximately 10.24 Gb of data was generated for each individual, and the average depth was 9.27X. High-quality reads were aligned to the chicken reference genome (GRCg6a), obtaining approximately 103,461,048 single-nucleotide polymorphism loci (SNPS).

### 3.1. LD Decay

This study examined linkage disequilibrium (LD) decay patterns across different chicken populations by estimating pairwise r^2^ values for SNPs within 300 kb genomic windows. The results demonstrated that LD decayed most rapidly in the BCY and TB populations (Figure 2), reflecting reduced linkage and potentially higher genetic diversity, possibly due to weaker selective pressures. In contrast, HYBC and the three commercial chicken breeds exhibited the slowest LD decay, suggesting stronger genomic linkage, which may result from intense artificial selection. The remaining local breeds displayed intermediate LD decay rates, indicative of moderate selective pressures and comparatively reduced genetic diversity.

### 3.2. Effective Population Size of Ancestors and Divergence Time

This study employed whole-genome SNPs and the SMC++ method to reconstruct *Ne* trajectories of 16 chicken breeds (Figure 3). Analysis shows that all the studied species diverged from a common ancestor approximately 100,000 years ago and, subsequently, the population experienced two expansion periods and two bottleneck periods.

The BCY population exhibited a contemporary *Ne* of 46,066 with a declining trajectory, suggesting recent demographic contraction potentially attributable to bottleneck effects or genetic drift. Nine indigenous breeds (JNBR, BY, LD, LX, LY, NY, TLF, YY, and XH) displayed congruent demographic patterns. In contrast, the Hetian Black Chicken and Turpan Gamecock populations from Xinjiang demonstrated sustained expansion trends, with contemporary *Ne* estimates of 321,212 and 191,077, respectively.

This study constructed three demographic history models and employed the Akaike Information Criterion (AIC) to evaluate and identify the best-fitting model (likelihood value: −3,202,912.453; AIC value: 211,256.901) for inferring the origin and divergence time of Xinjiang local chicken breeds (Figure 4A). Fastsimcoal analysis revealed that the divergence time between Baicheng Oli Chicken and Hetian Black Chicken occurred approximately 428–548 years ago, while Turpan Gamecock showed an earlier divergence time of about 894–943 years ago (Figure 4B).

### 3.3. Gene Flow Between Populations

Using Treemix software (v1.13) with RJF as the outgroup, we reconstructed a maximum-likelihood phylogent to assess gene flow and evolutionary relationships between BCY and other local/commercial chicken breeds (Figure 5). The analysis revealed significant admixture events, with distinct patterns of genetic introgression among populations: five migration events were detected between LH and local populations (BCY, NY, JNBR, LD, XH); four migration events were identified between RIR and populations (BCY, NY, JNBR, and YY); and no detectable gene flow was observed between WRR and any local chicken breeds. Additionally, gene flow was observed among some local breeds: one directional gene flow event from TLF to BCY; and two migration events involving TB, with genetic contributions to both BCY and NY populations.

To validate the putative gene flow events identified in phylogenetic analysis, we performed an ABBA-BABA test (D-statistics) under the population configuration (P1, P2, P3, Outgroup), where P1 and P2 represent local breeds, P3 denotes a commercial breed, and the outgroup is RJF (Figure 6). The key findings include that the ((NY, BCY), LH) combination exhibited the smallest |D| value (0.0150313), suggesting minimal gene flow from LH into BCY; the ((BY, NY), LH) test yielded the highest |D| value (0.098049), indicating significant introgression from LH into NY and BY; additional introgression events occurred, in that significant D-statistics supported gene flow YY, XH, TB and LY into either NY or BCY, corroborating the Tree Mix results; and no detectable introgression (|D| ≈ 0) was observed from RIR or WRR into BCY and NY.

This study utilized Dsuite’s Dinvestigate module to analyze trios exhibiting significantly elevated D-statistics, thereby refining and validating putative gene introgression events (Table 1). The analysis was performed using a sliding window approach with a window size of 2500 SNPs and a step size of 500 SNPs, during which we computed population genetic parameters (D, f_d, and f_dM). Genomic windows ranking in the top 1% of D-statistic values were identified as candidate introgression regions and subsequently subjected to functional annotation. The results demonstrated distinct patterns of introgression: the commercial breed LH exhibited predominant introgression into BCY populations localized to chromosomes GGA17, GGA22, GGA1, and GGA2, whereas Tibetan chicken populations showed primary introgression into BCY restricted to chromosome GGA5.

### 3.4. Gene Annotation of the BCY Infiltration Area by LH

The identified genes are functionally enriched in critical biological processes encompassing molecular fiber composition regulation, skeletal system development (including skeletal and striated muscle formation), muscle cell/tissue differentiation, embryonic development, reproductive system morphogenesis, limb development, and angiogenesis, with KEGG pathway analysis highlighting the pivotal involvement of Wnt and TGF-β signaling pathways in embryogenesis, folliculogenesis, and skeletogenesis, while further characterization revealed functionally categorized introgressed candidate genes including myogenic regulators (*MYF5*, *MYF6*), growth modulators (*FGFR1*, *TBX20*, *INHBA*), cell fate determinants (*BCL2*, *SOX10*), developmental mediators (*TWSG1*, *KITLG*), and RNA processing factors (*RBM24*). These are all associated with skeletal–muscular development, organogenesis, and reproductive traits (Figure 7).

This study investigated individual variation in introgressed genes and their potential functional mechanisms in Baicheng Oil Chicken. By calculating population differentiation indices (F*_ST_*) between BCY individuals and LH chickens across introgressed regions on chromosomes GGA1, GGA17, GGA2, and GGA22, we identified seven genomic regions with F*_ST_* = 0, containing five candidate genes: *RASSF3*, *RARX*, *CNTNAP2*, *CTNNAL1*, and ENSGALT00000104496. Functional annotation revealed that these genes are associated with key production traits and genetic characteristics of BCY chickens (Table 2).

Previous studies have shown that *RARX* is implicated in adipogenesis (fat deposition); *CTNNAL1* has documented associations with litter size in swine; *RASSF3* plays roles in immune cell infiltration; and *CNTNAP2* influences neural development and social behavior. These introgressed genes likely contribute to phenotypic variation in BCY chickens, particularly affecting metabolic, reproductive, immunological, and behavioral traits. The conserved regions (F*_ST_* = 0) suggest strong selective pressures maintaining these advantageous alleles in the BCY population (Table 2).

To further elucidate the introgression patterns of *CTNNAL1* and *RARX* genes within the BCY population, we constructed maximum likelihood trees using IQ-TREE software (v3.0) for the introgressed genomic regions containing these genes (Figure 8). Phylogenetic analysis revealed distinct clustering patterns: In the *RXRA* gene tree, 13 BCY individuals clustered with LH chickens. The *CTNNAL1* gene tree showed 25 BCY specimens forming a monophyletic clade with LH. The differences in FST values of the *CTNNAL1* and *RARX* regions and the structure of the gene trees indicate that there are differences in the degree of gene introgression among individuals within the BCY population.

To further elucidate the mechanisms by which introgressed genes affect traits, genetic variations in the *RXRA* and *CTNNAL1* genes were investigated. A single-nucleotide polymorphism in the sixth intron of the RXRA gene (7646507 C > T, rs316722310) was identified as a potential causative mutation. This site exhibits variation among individuals of the BCY population, with 3 individuals being heterozygous (3/30) and 27 being wild-type homozygous (27/30). In the *CTNNAL1* gene, four single-nucleotide variants were detected (88539016, 88539018 G > A; 88504039 G > A; 88519514 A > G), all of which occur in introns and may be associated with lncRNA or small non-coding RNA functions. Among the 30 individuals, 13 (13/30) showed no changes at these four sites, while 10 (10/30) exhibited identical mutations at SNPs 88539016 and 88539018 (Figure 9).

## 4. Discussion

This study used the SMC++ method to estimate the effective population sizes of Baicheng Oil Chicken, Hotan Black Chicken, and Turpan Game Chicken as 46,066, 321,212, and 191,077, respectively. However, according to statistics from the China’s National Animal Husbandry Bureau, the population sizes of these breeds are much larger: 362,975 for BCY, 1,187,656 for NY, and 825 for TLF. The observed divergence between SMC++-inferred historical effective population size and empirical census population size suggests that Ne, derived from genetic drift estimates based on allele frequency variance and heterozygosity decay rates, more accurately reflects the breeding-scale population dynamics than does the absolute census count [10,14]. The rapid LD decay observed in BCY, coupled with the low genomic linkage degree and high genetic diversity, provides robust support for the validity of our introgression inferences. These results demonstrate that Baicheng Oil Chicken exhibit a significantly reduced effective population size compared to their census size, suggesting a historical population bottleneck. This genomic signature underscores the necessity of implementing targeted conservation strategies to mitigate inbreeding depression and maintain genetic diversity in this breed [15,16].

Fsc2 is a highly flexible coalescent-based simulator framework capable of modeling complex evolutionary scenarios while enabling robust demographic parameter inference from the site frequency spectrum. This method has been extensively utilized across diverse taxa, including humans [17], animals [18], plants [19], and microbes [20], and supports the integration of ancient DNA data to elucidate temporal relationships between ancestral and modern populations.

In this study, we employed fsc2 to estimate divergence times among indigenous chicken breeds from Xinjiang. Our analyses indicate that Turpan gamefowl underwent domestication approximately 894–943 years before present (YBP), whereas Baicheng Oil Chicken and Hetian Black Chicken diverged 428–548 YBP. These estimates align with historical records documenting the cultural significance of gamefowl, which date back to at least 516 BCE in East Asia. Notably, Japanese gamefowl lineages were established prior to the Heian period (794–1192 CE), and historical accounts suggest that following the collapse of the Qing Dynasty, gamefowl populations expanded from Kaifeng to Zhengzhou, Luoyang, Shandong, and ultimately into Xinjiang [21]. The concordance between our genomic estimates and documented historical dispersal patterns strongly supports the inferred divergence chronology of Turpan gamefowl.

Recent genomic studies have shed light on the divergence timelines of various indigenous chicken populations. Shourong Shi et al. [22] reported that Tibetan chickens likely diverged approximately 1000 YBP, while Lindian chickens diverged 600–800 YBP, consistent with historical documentation in the “Lingnan Zashi” poetic records [23]. Similarly, historical archives indicate that Bacheng Oil Chickens have been selectively bred for over 300 years [24], and Hetian Black Chickens were widely reared during the Jingjue Kingdom period (~1750 YBP) [25].

Our evolutionary phylogenomic analyses, based on whole-genome sequencing data, estimate that divergence between Baicheng Oil Chicken and Hetian Black Chicken occurred more recently than the differentiation of Turpan gamefowl, a timeframe strongly supported by both historical records and molecular dating [8]. To our knowledge, this study represents the first comprehensive genomic investigation of the population structure, domestication history, and divergence chronology of indigenous chicken breeds in Xinjiang. These findings provide a robust genomic framework for reconstructing the evolutionary history of Xinjiang’s local poultry populations.

In addition, this study identified significant gene flow from commercial LH chickens into Baicheng Oil Chicken, corroborating previous hypotheses regarding widespread genomic introgression from commercial breeds into indigenous Chinese chicken populations [26].

Our analysis identified 1291 protein-coding genes within introgressed regions from LH chicken to Baicheng Oil Chicken. Functional annotation revealed that these candidate genes are significantly enriched for biological processes including skeletal morphogenesis, muscle tissue development, organ growth regulation and reproductive system function. These genes predominantly participate in two key developmental pathways: the Wnt signaling pathway and TGF-β signaling pathway. The functional profile suggests these introgressed loci may contribute to embryonic pattering and development, folliculogenesis and gametogenesis, osteogenesis and skeletal formation [27,28]. These findings provide the first genomic evidence that LH-derived introgression events may have influenced growth and reproductive performance traits in Baicheng Oil Chicken. Notably, this represents a novel discovery in avian genomics, as no comparable reports of such introgression patterns have been previously documented in the literature.

Egg production in poultry represents a complex sex-limited quantitative trait influenced by polygenic inheritance patterns, environmental interactions, and management practices. Given its characteristically low heritability (h^2^ ≈ 0.2–0.3), conventional phenotypic selection approaches demonstrate limited efficacy for genetic improvement [29].

Four introgressed single-nucleotide variants were identified (88539016, 88539018 G > A; 88504039 G > A; 88519514 A > G) within the *CTNNAL1* gene. All identified variants occur in non-coding intronic regions, suggesting potential regulatory impacts on mRNA splicing efficiency, transcript stability, and gene expression modulation [30,31]. The variants originate from LH chicken introgression events, indicating the possible transmission of production-related genetic elements. Notably, CTNNAL1 (α-catulin) plays established roles in cytoskeletal organization, cell adhesion processes, and ovarian follicle development. These findings provide novel insights into the genetic architecture underlying egg production traits in indigenous chicken breeds.

Previous studies using the candidate gene approach to identify potential genes affecting litter size in pigs have found that the *CTNNAL1* gene locus c.1878 G > C is associated with the number of live piglets in Large White pigs. The CC mutation homozygote has 1.14 more live piglets than the GG wild type (*p* < 0.01), and the differences in total and live piglets in Chinese DIV pigs are 2.62 and 2.07, respectively (*p* < 0.01) [32]. The CG heterozygote has more live piglets than the GG and CC types, with dominant effects of 0.25 and 0.10, respectively (*p* < 0.05) [33]. Additionally, studies have found that *CTNNAL1* gene mRNA is expressed in the human testes, ovaries [34], skeletal muscle, and placenta [35]. Therefore, this study speculates that the *CTNNAL1* gene may be a favorable gene for egg production.

*RXRA* gene, located on chicken chromosome 17 (764–765 Mb), encodes a key member of the nuclear receptor superfamily that regulates multiple metabolic pathways, including lipid homeostasis, glucose metabolism, energy balance, and hormone signaling [36,37]. In poultry, lipid metabolism is a crucial factor affecting growth traits during the induction of preadipocytes into mature adipocytes. Previous studies have shown that the *RXRA* gene in cattle is located on chromosome 1 (105.98–106.01 Mb), and its quantitative trait loci (QTLs) are associated with growth performance and meat quality [38]. These findings suggest that the *RXRA* gene may influence lipid metabolism and meat quality attributes.

To sum up, these results underscore the pressing need to implement conservation measures that safeguard China’s indigenous chicken breeds from progressive genetic dilution. The preservation of these unique genetic resources is essential for maintaining agricultural biodiversity and ensuring future breeding options in the face of changing environmental and market demands.

## 5. Conclusions

As one of the local chicken breeds in Xinjiang, the population LD of Baicheng Oil Chicken has declined rapidly, and the historical effective population is relatively small. The differentiation was between approximately 428 and 548. Moreover, commercial breed LH may have an infiltrated into Baicheng Oil Chicken, mainly in GGA17, GGA1 and GGA2, accounting for approximately 7%. The infiltrated gene CTNNAL1 may have an impact on its egg production, while the RARX gene may play a role in its fat formation process.

## Figures and Tables

**Figure 1 animals-15-01952-f001:**
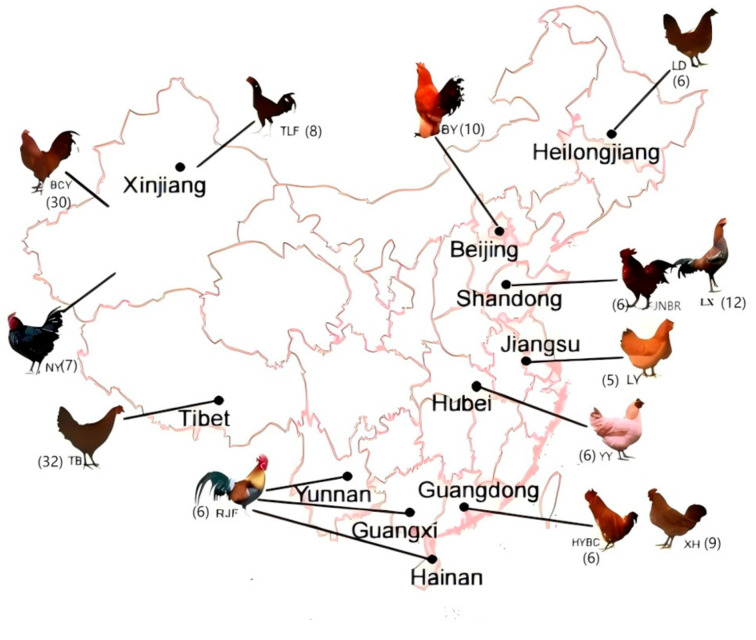
The geographical location of local chicken breed samples.

**Figure 2 animals-15-01952-f002:**
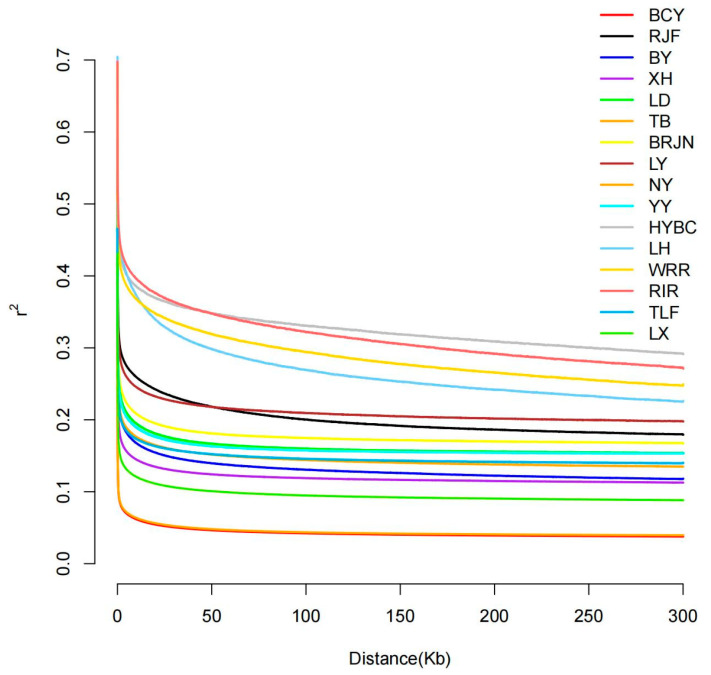
LD decay of the 16 chicken breeds.

**Figure 3 animals-15-01952-f003:**
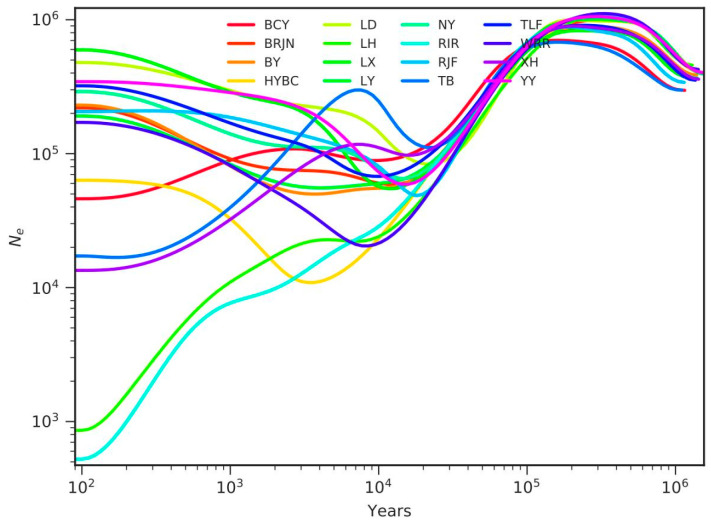
Ancestral historical effective population size. Note: *Ne* = effective population size.

**Figure 4 animals-15-01952-f004:**
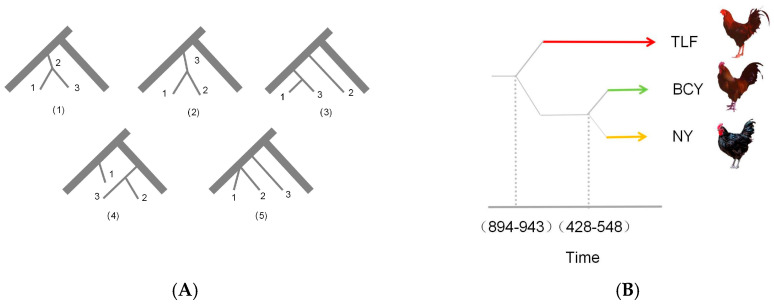
Population historical dynamics. Note: RJF is the outgroup, with 1 representing BCY population, 2 representing NY population, and 3 representing TLF population. (**A**) Possible historical model; (**B**) the differentiation times of the three populations.

**Figure 5 animals-15-01952-f005:**
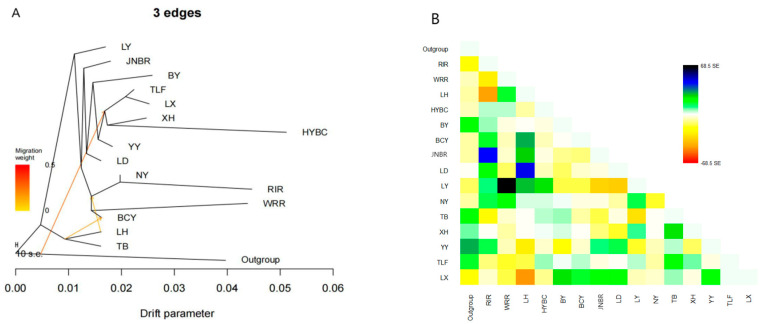
Genetic migration events of local chickens in Xinjiang from commercial chickens or other local chickens (M = 3, M = 12). Note: (**A**,**C**) Tree Mix diagram. Arrows from one branch of the tree to another, indicating gene flow events. Migration weight is indicated according to arrow color. (**B**,**D**) Residual graph representing the fit of the model. The color indicates the difference between the actual data and the model’s predictions.

**Figure 6 animals-15-01952-f006:**
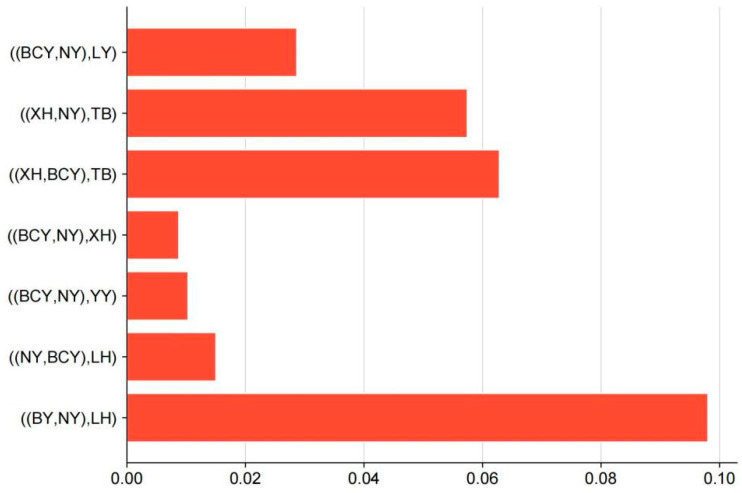
D-statistic for introgression between commercial chicken or other local chicken and local chicken of Xinjiang.

**Figure 7 animals-15-01952-f007:**
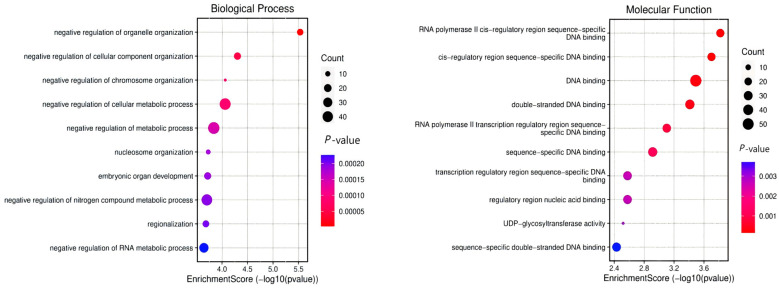
GO and KEGG analysis of infiltrated candidate genes from LH to BCY.

**Figure 8 animals-15-01952-f008:**
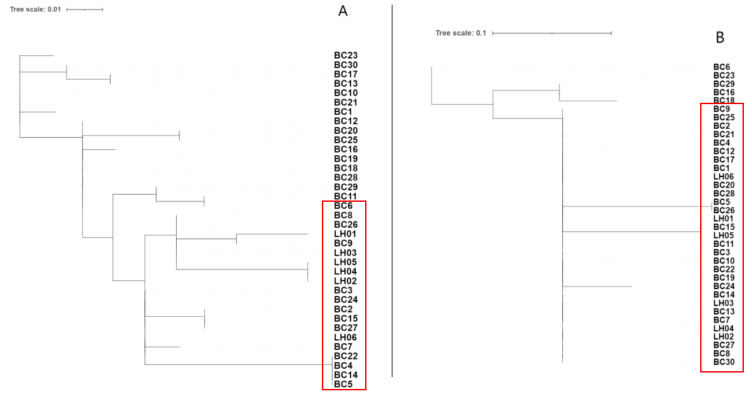
ML tree constructed using introgression gene sequences. Note: (A) ML tree of *RXRA* (chr17:7577223-7651643) gene. (B) ML tree of *CTNNAL1* (chr2:88491627-88549212) gene.

**Figure 9 animals-15-01952-f009:**
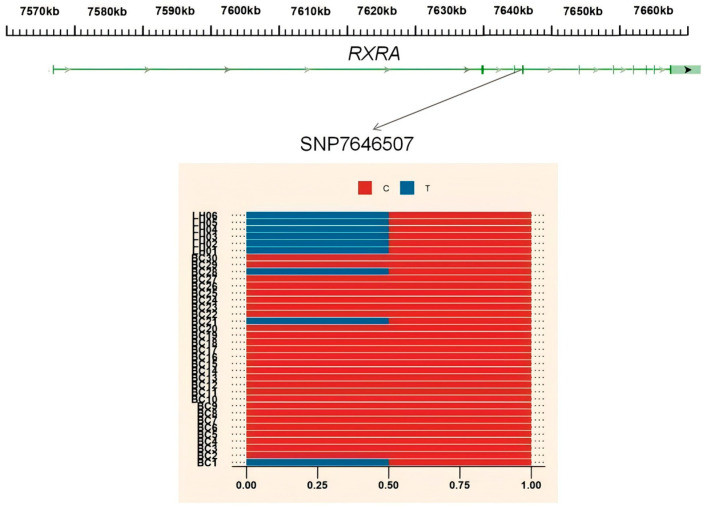
Allele frequencies of two SNPs in intron 1 of the *RXRA* and *CTNNAL1* gene in BCY and LH chicken populations.

**Table 1 animals-15-01952-t001:** The top 1% of introgression regions of the chromosome.

((P1,P2),P3)	chr	Window Start	Window End	D Value	f_d	f_dM	d_f
((NY,BCY),LH)	chr17	6738123	9797958	0.074077	0.060236	0.042896	0.039979
	chr22	2015588	4127909	0.0738	0.063796	0.044762	0.037595
	chr1	20420138	53163584	0.070559	0.063676	0.042755	0.038425
	chr2	47984993	92026288	0.069768	0.044631	0.034249	0.037573
	chr2	58104921	100875426	0.069497	0.052234	0.038317	0.035799
((XH,BCY),TB)	chr5	24005918	49859805	0.141173	0.23659	0.114761	0.078218
	chr5	16171926	38768130	0.137803	0.236848	0.111738	0.078419

**Table 2 animals-15-01952-t002:** Chromosome positions and genes with zero *F_ST_.*

CHROM	BIN_START	BIN_END	GENE
chr1	33820001	33830000	RASSF3
chr17	7640001	7650000	RXRA
chr2	49130001	49140000	No gene
chr2	54155001	54165000	CNTNAP2
chr2	88495001	88555000	CTNNAL1
chr2	88550001	88560000	FAM206A
chr2	88625001	88650000	ENSGALT00000104496

## Data Availability

The original contributions presented in this study are included in the article. Further inquiries can be directed to the corresponding author(s).

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
