# Peer review of "Research on the Historical Dynamics of Baicheng Oil Chicken Populations"

_animals, 2025, doi:10.3390/ani15131952_

Round 1
Reviewer 1 Report
Comments and Suggestions for Authors
Based on the whole gene resequencing data of 16 chicken breeds, this study analyzed the historical dynamics of the population by means of population genetics. The results will provide a theoretical basis for the scientific protection and utilization of Baicheng Fatty Chicken germplasm resources. The research ideas are clear and the research results are rich. However, there are still some problems, and it is recommended to make some revisions.
- Line 46, the effective population size (Ne) belongs to the parameter of population genetics research. Population genetics should be put forward before Ne.
- The number of individuals of each chicken breeds can be supplemented in Figure 1.
- Lines 72-72 indicates that the sequencing depth is at least 10×, but the average sequencing depth in the 117th line is 9.27×.
- Line 134, the full name of Ne has been marked above, and the abbreviation can be used here.
- In line 105, P should be italicized.
- Figure 6 should be labeled with the adjusted P-value.
- Please improve the clarity of the Figure.
Author Response
- Line 46, the effective population size (Ne) belongs to the parameter of population genetics research. Population genetics should be put forward before Ne.
Response:Thank you for pointing this out. Therefore, We have made the necessary changes. See line 47.
- The number of individuals of each chicken breeds can be supplemented in Figure 1.
Response:Thank you for pointing this out. We agree with this comment. Therefore,we have supplemented the number of individuals of each chicken breeds in Figure 1. See line 75.
- Lines 72-72 indicates that the sequencing depth is at least 10×, but the average sequencing depth in the 117th line is 9.27×.
Response:Thank you for pointing this out. Lines 74 indicates that sequencing depth of raw reads is at least 10X,but lines 117-120 indicates that the average depth of clean reads which has been quality controlled by raw reads. So, the data are inconsistent. Therefore, We have made the necessary changes. See line 118.
- Line 134, the full name of Ne has been marked above, and the abbreviation can be used here.
Response:Thank you for pointing this out. We agree with this comment. Therefore, we have deleted the historical effective population size. See line 135. And we have made uniform and revisions throughout the entire text.
- In line 105, P should be italicized.
Response:Thank you for pointing this out. We agree with this comment. Therefore, we have made the modification See line 105.
- Figure 6 should be labeled with the adjusted P-value.
Response:Thank you for pointing this out. We agree with this comment. Therefore, we have made the modification. See Figure 7.
- Please improve the clarity of the Figure.
Response:Thank you for pointing this out. We agree with this comment.
Therefore, we have made the modification. See Figure1,Figure 2,Figure 3, Figure 4.
Reviewer 2 Report
Comments and Suggestions for Authors
- The authors provide data on the number of sequenced samples in geographical locations of chicken studies.
- There is no citation to the article from which the information is provided! (lines 225-227)
Author Response
- Theauthors provide data on the number of sequenced samples in geographical locations of chicken
Response:Thank you for pointing this out. We agree with this comment. Therefore,we have supplemented the number of individuals of each chicken breeds in Figure 1.
- There isno citation to the article from which the information is provided! (lines 225-227)
Response:Thank you for pointing this out. This result is the outcome of our experiment, so no references are presented. We cited the literature in our discussion. See line 335-353.
Author Response
The content of this opinion is inconsistent with my research content.
Round 2
Reviewer 3 Report
Comments and Suggestions for Authors
The authors addressed my questions and comments very vell.